# A Newly Developed Method-Based Xanthine Oxidoreductase Activities in Various Human Liver Diseases

**DOI:** 10.3390/biomedicines11051445

**Published:** 2023-05-14

**Authors:** Ken Sato, Atsushi Naganuma, Tamon Nagashima, Yosuke Arai, Yuka Mikami, Yuka Nakajima, Yuki Kanayama, Tatsuma Murakami, Sanae Uehara, Daisuke Uehara, Yuichi Yamazaki, Takayo Murase, Takashi Nakamura, Toshio Uraoka

**Affiliations:** 1Department of Gastroenterology and Hepatology, Gunma University Graduate School of Medicine, Maebashi 371-8511, Japan; 2Department of Hepatology, Heisei Hidaka Clinic, Takasaki 371-0001, Japan; 3Department of Healthcare Informatics, Takasaki University of Health and Welfare, Takasaki 370-0033, Japan; 4Department of Gastroenterology, National Hospital Organization Takasaki General Medical Center, Takasaki 370-0829, Japan; 5Department of Gastroenterology, National Hospital Organization Shibukawa Medical Center, Shibukawa 377-0204, Japan; 6Mie Research Park, Sanwa Kagaku Kenkyusho, Inabe 511-0406, Japan

**Keywords:** xanthine oxidoreductase, liver disease, etiology, alanine aminotransferase, xanthine, oxidative stress, inflammation

## Abstract

Studies evaluating xanthine oxidoreductase (XOR) activities in comprehensive liver diseases are scarce, and different etiologies have previously been combined in groups for comparison. To accurately evaluate XOR activities in liver diseases, the plasma XOR activities in etiology-based comprehensive liver diseases were measured using a novel, sensitive, and accurate assay that is a combination of liquid chromatography and triple quadrupole mass spectrometry to detect [^13^C_2_, ^15^N_2_]uric acid using [^13^C_2_, ^15^N_2_]xanthine as a substrate. We also mainly evaluated the association between the plasma XOR activities and parameters of liver tests, purine metabolism-associated markers, oxidative stress markers, and an inflammation marker. In total, 329 patients and 32 controls were enrolled in our study. Plasma XOR activities were generally increased in liver diseases, especially in the active phase, such as in patients with hepatitis C virus RNA positivity, those with abnormal alanine transaminase (ALT) levels in autoimmune liver diseases, and uncured hepatocellular carcinoma patients. Plasma XOR activities were numerically highest in patients with acute hepatitis B. Plasma XOR activities were closely correlated with parameters of liver tests, especially serum ALT levels, regardless of etiology and plasma xanthine levels. Our results indicated that plasma XOR activity might reflect the active phase in various liver diseases.

## 1. Introduction

Xanthine oxidoreductase (XOR) is an enzyme that is regulated on multiple levels, which is widely distributed in living organisms and plays many physiological roles [1]. The principal XOR activity includes xanthine dehydrogenase (XDH) activity and xanthine oxidase (XO) activity [1]. XOR catalyzes the conversions of hypoxanthine to xanthine and of xanthine to uric acid, which are the last two steps of purine catabolism [1]. XO generates reactive oxygen species (ROS) [1]. In mammalian organs, XOR is predominantly XDH, which can be converted into XO either in an irreversible or a reversible fashion [2,3,4]. XDH exists in several tissues, such as the liver and intestine, leaks into the bloodstream and is there converted into XO [5,6]. Thus, XOR is considered a source of ROS generation and consequently contributes to the occurrence of oxidative stress-associated tissue damage [7,8,9].

Elevation of serum or liver XO or XOR has been reported in human liver diseases and experimental liver injury animal models [10,11]. However, in these studies [10,11] analyzing comprehensive liver diseases, the number of study subjects in each disease group was small, and the methodology for the measurement of XO or XOR seems to be outdated.

The accurate measurement of XOR activity is difficult because XOR activity is very low in humans compared to rodents [12,13,14]. Therefore, we used a combination of liquid chromatography and triple quadrupole mass spectrometry (LC/TQMS) to detect [^13^C_2_, ^15^N_2_]uric acid using [^13^C_2_, ^15^N_2_]xanthine as a substrate, which has recently been established as a novel, sensitive, and accurate assay for plasma XOR activity in humans [14]. In the current study, we comprehensively analyzed human plasma XOR activities in various liver diseases and mainly evaluated the association between XOR activities and parameters of liver tests, purine metabolism-associated markers, oxidative stress markers, and an inflammatory marker.

## 2. Materials and Methods

The present study is a prospective multicenter cross-sectional clinical study performed in accordance with the Declaration of Helsinki and the Ethical Guidelines for Medical and Health Research Involving Human Subjects and was approved by the ethics committee of each institution. All patients signed written informed consent forms prior to enrollment.

### 2.1. Study Population

The present study was performed in Gunma University Hospital or affiliated hospitals between February 2018 and May 2022, with the enrollment of patients with acute or chronic liver diseases or controls with normal liver tests confirmed by check-ups within the past 12 months or laboratory tests and aged 20 years old or older. The diagnosis of each liver disease was based on the judgment of the attending physicians who were hepatologists. The exclusion criteria were as follows: (1) patients who had taken XO inhibitors within the past 4 weeks because plasma XOR activity cannot be detected due to the XO inhibitors; (2) patients with acute infection such as dental caries or active extrahepatic malignancy; and (3) patients who were not considered eligible for the study based on the doctor’s assessment, such as patients with severe kidney disease, collagenous diseases with high values of high-sensitivity C-reactive protein (hs-CRP), or those without a postprandial blood sample available. Patients with more than one liver disease were excluded when we could not exclude the effects of multiple etiologies on the pathophysiology of liver diseases except overlap syndrome, which is a combined autoimmune liver disease. The subjects included some of the patients who were in treatment for their original liver disease to reflect real-world data.

### 2.2. Blood Sample Collection and Preparation

Vital signs, height, and body weight were measured on the same day as the blood collection date in principle. Blood samples were collected in the morning after overnight fasting from controls and patients. Plasma or serum aliquots that were obtained after separation of the blood samples by centrifugation were collected and kept frozen at −80 °C until subsequent analyses. We used plasma samples for the measurement of XOR, xanthine, hypoxanthine and uric acid, and serum samples for the measurement of biochemical markers, including hs-CRP and oxidative stress markers. In addition, we performed an investigation regarding current complications such as hypertension and cancer, past history of diseases, and history of taking medicines that cause secondary hyperuricemia within the past 4 weeks.

### 2.3. Measurement of the Hypoxanthine and Xanthine Concentrations

We measured plasma concentrations of xanthine and hypoxanthine using LC/TQMS (Nexera/QTRAP 4500; SHIMAZU, Kyoto, Japan/SCIEX, Framingham, MA, USA) as previously described [15]. In brief, the plasma samples obtained after separation by centrifugation were used, and methanol containing [^13^C_2_, ^15^N_2_]xanthine and [^13^C_2_, ^15^N_2_]hypoxanthine was added as internal standards and subsequently subjected to centrifugal treatment at 3000 × *g* at 4 °C for 15 min. Distilled water was used to dilute the supernatant by a factor of 5. The concentrations of xanthine and hypoxanthine were measured by LC/TQMS (SHIMAZU Nexera/SCIEX QTRAP 4500).

### 2.4. Measurement of Plasma XOR Activity

Low molecular compounds in plasma samples were removed by Sephadex G-25 column before measurement of plasma XOR activity. We measured plasma XOR activity by a combination of LC/TQMS to detect [^13^C_2_, ^15^N_2_]uric acid using [^13^C_2_, ^15^N_2_]xanthine as a substrate as previously described [14,16]. The LC/TQMS consists of a Nexera LC system (SHIMADZU., Kyoto, Japan) and a QTRAP4500 (SCIEX, Framingham, MA, USA) equipped with an ESI interface. We also measured calibration standards of [^13^C_2_, ^15^N_2_]uric acid and identified the amount of production based on the calibration curve. XOR activities are given as pmol/h/mL [12,13,14].

### 2.5. Measurement of Serum Oxidative Stress Markers

The oxidative stress in serum samples was evaluated based on both oxidant and antioxidant components. The levels of derivatives of reactive oxygen metabolites (d-ROMs), which are biomarkers for quantification of the oxidative status through the evaluation of hydroperoxidation of organic compounds, were measured by the d-ROM test. The biological antioxidant potential (BAP) and total antioxidant barrier, which are other biomarkers reflecting the antioxidant capacity, were measured by the BAP test and oxy-absorbent test, respectively. These markers were measured using a free radical elective evaluator system (FREE CARRIO DUO, Wismerll Co., Ltd., Tokyo, Japan). Briefly, the frozen serum was thawed, and 20-μL serum samples were added to a 1.2-mL assay mixture (acetic acid buffer solution) in a prewarmed cuvette. Then, the resulting solution was gently mixed for 5 to 10 s. Next, 20 μL of aromatic compound as a chromogenic substrate was added and gently mixed for 5 to 10 s. The cuvette was then incubated in a thermostatic block for 5 min at 37 °C. Thereafter, the absorbance at 505/546 nm was recorded, and the d-ROMs were calculated. The measurements were expressed as U. CARR. For the BAP test, 50 μL of iron(III) chloride as a chromogenic substrate was added to a prewarmed cuvette that contained thiocyanate and was gently mixed for 5 to 10 s. Then, the cuvette was incubated for 5 min in a thermostatic block, and the first absorbance at 505 nm was measured. After removing the cuvette from the block, 10 μL of the thawed serum samples was added and gently mixed for 5 to 10 s. Next, the cuvette was incubated for 5 min in the thermostatic block, and the second absorbance at 505 nm was measured. The concentration of BAP was calculated. The measurements are expressed as μmol/L. Regarding the oxy-absorbent test, 10 μL of the thawed serum samples was diluted 1:100 with distilled water. A total of 1000 μL of hypochlorous acid (HOCl) solution as a chromogenic substrate was added to cuvettes. Then, 10 μL of diluted serum sample or distilled water as a blank sample was added to cuvettes containing HOCl, gently mixed for 5 to 10 s and incubated for 10 min at 37 °C. Then, 10 μL of aromatic compound as a chromogenic substrate used for reactive oxygen metabolite determination was added. The intensity of the complex, which is inversely associated with the antioxidant power, was measured at 505/546 nm, and the total antioxidant barrier was calculated. The measurements are expressed as μmol/mL.

### 2.6. Statistical Analyses

Numeric values are shown as the medians (interquartile ranges), unless otherwise stated. Comparisons between two groups were analyzed using the Mann‒Whitney U test. The Kruskal‒Wallis test and Bonferroni’s multiple comparison test as a post hoc test were performed to test significant differences in variables among multiple groups. Groups with at least 3 or more subjects were judged to be comparable. Comparisons of groups composed of the same etiology were performed after exclusion of patients with hepatocellular carcinoma (HCC), who were separately analyzed because the number of patients with HCC was relatively small. The correlation between two values was analyzed using Spearman’s rank correlation coefficient. All statistical analyses were carried out by using IBM SPSS statistics 26 (IBM, Tokyo, Japan), with *p* < 0.05 indicating statistical significance. Plasma XOR levels greater than the upper limit of quantification were not used for the correlation analyses. hs-CRP, xanthine, and hypoxanthine levels less than the lower limit of quantification were not used for comparative or correlation analyses. A total of 329 patients and 32 controls were enrolled in the present study.

## 3. Results

### 3.1. Baseline Characteristics of Patients and Controls

The baseline characteristics of the patients (144 males/185 females) and controls (8 males/24 females) enrolled in the present study are shown in Table 1. The median ages of the patients and controls were 67 and 39, respectively. The etiologies of liver diseases were chronic hepatitis C with or without human immunodeficiency virus (HIV) infection, chronic hepatitis B with or without past hepatitis C virus (HCV) infection, acute hepatitis B, non-B non-C liver cirrhosis, nonalcoholic fatty liver disease (NAFLD), alcoholic liver disease (ALD), autoimmune hepatitis (AIH), primary biliary cholangitis (PBC), overlap syndrome (AIH + PBC), and overlap syndrome [AIH + primary sclerosing cholangitis (PSC)]. The stages of liver diseases included acute hepatitis, chronic liver disease without cirrhosis (CH), liver cirrhosis (LC), and HCC. No patients received liver transplantation. The numbers of subjects complicated with diabetes mellitus, hypertension, hyperuricemia, and hyperlipidemia were 77, 156, 32, and 119, respectively.

### 3.2. Plasma XOR Activities Based on Liver Disease Etiologies

Plasma XOR activities in each etiology are shown in Table 2 and were compared with those in controls. A significant difference compared to the controls was obtained in patients with chronic hepatitis C, chronic hepatitis B, NAFLD, ALD, PBC, AIH, and overlap syndrome (AIH + PBC and AIH + PSC). Notably, plasma XOR activity was quite high (more than 6670 pmol/h/mL) in acute hepatitis B, although the number of patients was only one.

### 3.3. Plasma XOR Activities in Viral Hepatitis

Regarding the viral hepatitis groups, significant differences among controls and the other groups were obtained and are shown in Figure 1A–C. Patients with chronic hepatitis C comprised those with positivity for serum HCV RNA and those with negativity for serum HCV RNA. The former indicates current active infection of HCV, and the latter indicates past infection of HCV that was treated by antivirals in our study subjects. The plasma XOR activities in patients with positivity for serum HCV RNA were significantly higher than those of controls or those with negativity for serum HCV RNA (Figure 1A). Regarding the chronic hepatitis C stage, the plasma XOR activities in patients with CH and serum HCV RNA positivity were significantly higher than those in controls (Figure 1B). The plasma XOR activities in patients with liver cirrhosis with serum HCV RNA positivity had a trend toward increasing values compared to those in controls (*p* = 0.051). In the patients with negative serum HCV RNA, no significant differences among the control, CH, and LC groups were observed according to a post hoc test (data not shown). Regarding the chronic hepatitis B stage, HBV carriers (inactive carriers and asymptomatic carriers) whose transaminase levels are normal without any treatment were separately analyzed from CH. The number of patients with LC in the groups did not meet the comparison criteria. The plasma XOR activities in patients with CH but not HBV carriers were significantly higher than those of controls (Figure 1C).

### 3.4. Plasma XOR Activities in ALD and NAFLD

Regarding the ALD and NAFLD groups, significant differences among controls and the other groups were obtained and are shown in Figure 2A–E. The results of the post hoc test are described as follows. For patients with ALD, the plasma XOR activities in patients with CH but not those with LC were significantly higher than those of controls (Figure 2A). Serum ALT and γGTP levels reflect the disease activity in ALD patients, and the plasma XOR activities were separately analyzed with respect to serum ALT or γGTP levels. The number of patients with LC in the groups did not meet the comparison criteria. Regarding CH in ALD patients, the plasma XOR activities in patients with abnormal ALT values were significantly higher than those of controls (Figure 2B). There were no significant differences in the plasma XOR activity among controls, those with abnormal γGTP values, or those with normal γGTP values (data not shown).

Regarding the NAFLD stage, the plasma XOR activities in patients with CH but not those with LC were significantly higher than those of controls (Figure 2C). Serum ALT and γGTP levels reflect the disease activity in NAFLD patients, and the plasma XOR activities were separately analyzed with respect to serum ALT or γGTP levels. The number of patients with LC in the groups did not meet the comparison criteria. For NAFLD patients with CH, the plasma XOR activities in those with abnormal ALT values were significantly higher than those of controls or those with normal ALT values (Figure 2D). In the same population, the plasma XOR activities with abnormal or normal γGTP values were significantly higher than those of controls (Figure 2E).

### 3.5. Plasma XOR Activities in Autoimmune Liver Diseases

In the AIH and PBC groups, significant differences among the controls and the other groups were obtained and are shown in Figure 3A–E. The results of the post hoc test are described as follows. In the AIH stage, the plasma XOR activities in patients with CH and those with LC were significantly higher than those of controls (Figure 3A). Serum ALT levels reflect the disease activity in AIH patients, and the plasma XOR activities were separately analyzed based on the serum ALT levels. The number of patients with LC in the groups did not meet the comparison criteria. With respect to AIH patients with CH, the plasma XOR activities in patients with abnormal ALT levels were significantly higher than those of controls or those with normal ALT levels (Figure 3B).

 In patients with the PBC stage, the plasma XOR activities in patients with LC but not those with CH were significantly higher than those of controls (Figure 3C). Serum ALT, ALP, and γGTP levels reflect the disease activity in PBC patients, and the plasma XOR activities were separately analyzed based on the serum ALT, ALP, and γGTP levels. The number of patients with LC in the groups did not meet the comparison criteria. With respect to CH in PBC patients, the plasma XOR activities in those with abnormal ALT values were significantly higher than those of controls or those with normal ALT levels (Figure 3D). In the same population, the plasma XOR activities with abnormal ALP levels were significantly higher than those of controls (Figure 3E). There were no significant differences among controls, those with abnormal γGTP levels, or those with normal γGTP levels (data not shown).

### 3.6. Plasma XOR Activities in HCC

In HCC, the patients consisted of uncured (active) or cured patients. Significant differences among controls, uncured patients, and cured patients were obtained. The plasma XOR activities in the uncured patients were significantly higher than those in the controls or the cured patients according to the post hoc test (Figure 4).

### 3.7. The Association between Plasma XOR Activities and Parameters of Liver Tests

Significant positive correlations between plasma XOR activities and serum ALT, AST, ALP, or γGTP levels were obtained (Figure 5A,B). Numerically, the strongest linear relationship observed was between plasma XOR activities and serum ALT levels, with a correlation coefficient of 0.829 (Figure 5A). On the other hand, there were no significant correlations between plasma XOR activities and other parameters of liver tests, such as total bilirubin, albumin, or prothrombin time-international normalized ratio (Supplemental Appendix A).

### 3.8. The Association between Plasma XOR Activities and Purine Metabolism-Associated Markers, Oxidative Stress Markers and an Inflammation Marker

Significant positive correlations between plasma XOR activities and plasma xanthine levels (Figure 6A) or uric acid levels (Figure 6B) were obtained. Numerically, the strongest linear relationship was observed between plasma XOR activities and xanthine levels, with a correlation coefficient of 0.455. On the other hand, there were no significant correlations between plasma XOR activities and hypoxanthine levels (Supplemental Appendix A). Regarding oxidative stress markers, a significant positive correlation between plasma XOR activities and serum d-ROMs was obtained (Figure 6C). On the other hand, there were no significant correlations between plasma XOR activities and serum BAP or total antioxidant barrier (Supplemental Appendix A). Regarding the inflammation marker, a significant positive correlation between plasma XOR activities and serum hs-CRP values was obtained (Figure 6D).

### 3.9. The Association between Plasma XOR Activities and Body Mass Index (BMI)

Significant positive correlation between plasma XOR activities and BMI was obtained (Figure 7).

## 4. Discussion

The most important findings of our study are that plasma XOR activities were generally increased in liver diseases, especially in the active phase, and are strongly associated with liver transaminase levels, especially serum ALT levels, regardless of etiology. These results were supported by the findings that plasma XOR activity was very high in acute hepatitis B, in which serum ALT levels were quite high. The plasma XOR activities were closely associated with plasma xanthine levels, the d-ROM value that reflects oxidative stress, and hs-CRP levels.

We think that the simple comparison of the plasma XOR activities among different etiologies of liver diseases was difficult because plasma XOR activities were closely correlated with serum ALT levels. Plasma XOR activities were higher in the active phase than in the recovery phase due to the appropriate treatment intervention in the same etiology of liver diseases, such as chronic hepatitis C with positivity for HCV RNA, NAFLD, AIH, PBC, and HCC, but not ALD. With respect to liver disease staging, plasma XOR activities in patients with CH in various diseases were significantly higher than those of controls except for those with PBC, although the activity in patients with PBC had a trend toward increasing compared to those in controls (*p* = 0.053). On the other hand, plasma XOR activities in patients with LC in PBC and AIH were significantly higher than those in controls, and the activities in chronic hepatitis C patients with LC and serum HCV RNA positivity had a trend toward increasing compared to those in controls (*p* = 0.051). The possible reason is that serum ALT levels in patients with LC were significantly lower than those in patients with CH in NAFLD and had a trend toward decreasing compared to those of CH in ALD patients (*p* = 0.091). In addition, there were no significant differences in plasma XOR activities between patients with LC and those with CH in any etiology. Thus, plasma XOR activity may be dependent on serum ALT levels, not disease stage or etiology, in various liver diseases.

XOR is expressed in the cell, primarily in hepatocytes, and is also present in bile duct epithelial cells and is released into the extracellular matrix, plasma, or bile in cell turnover or cell damage [17,18]. In addition, hepatic XOR activity was 10- to 20-fold higher in patients with liver disease than in healthy controls [18]. ALT and AST are deviation enzymes, of which blood levels increase during conditions of hepatocyte damage or increased permeability of cell membranes. ALT is more specific to the liver than AST. In addition, as the half-life of ALT is longer than that of AST, ALT may reflect chronic liver injury more so than AST. ALP is present in numerous mammalian tissues, and serum ALP levels are mainly used as a marker of liver disease, especially cholestatic disease [19,20]. Many nonhepatic factors are involved in the increase in serum ALP levels [20]. ALT is regarded as more specific for hepatic injury than ALP. γGTP is a nearly ubiquitous epithelial enzyme that plays a role in the catabolism of extracellular glutathione, and its elevation is found in liver diseases such as ALD, NAFLD, and cholestatic liver disease [21]. γGTP is considered to be less specific for hepatic injury than ALT [21]. Thus, the plasma XOR activities were well correlated with parameters of liver tests, especially serum ALT levels.

As mentioned above, plasma XOR activity seems to reflect XOR protein levels that are released from hepatocytes and bile duct epithelial cells during liver damage to some extent. However, a part of XOR protein was not reflected in plasma XOR activity. The reason is that XOR is composed of three domains that is Fe-S, FAD, and molybdopterin domains [22] and molybdenum-containing enzyme can itself include a second inactive form that is desulpho-XOR, in which a Mo = S grouping is replaced by Mo = O [23]. In addition, plasma XOR activity may apparently decrease at measurement of the activity when endogenous hypoxanthine, xanthine, and uric acid exist in the assay system by their competitive inhibitions. To prevent these possibilities, the samples were purified on a Sephadex G-25 column in advance in our study as previously described [12,14].

Xanthine and uric acid are produced from hypoxanthine and xanthine by XDH in purine catabolism, respectively, as mentioned above [1]. Hypoxanthine that is derived from inosine is hydroxylated to produce xanthine. Xanthine is also made by deamination of guanine. Then, uric acid is formed by hydroxylation of xanthine in cytosol [24]. Thus, it makes sense that plasma XOR activities were significantly correlated with the plasma xanthine and uric acid levels in study subjects, including those with liver diseases and controls. However, plasma XOR activities were not significantly correlated with the plasma hypoxanthine levels. One of the plausible reasons is that the metabolism of hypoxanthine is quite different from that of xanthine or uric acid. As xanthine and hypoxanthine are precursors of uric acid and are oxidized by XDH [1,24], XOR inhibitors are considered to increase the levels of both precursors. An XOR inhibitor, however, significantly increased plasma xanthine levels but not hypoxanthine levels in Japanese male subjects [25]. There is a salvage pathway of purine metabolism, in which hypoxanthine-guanine phosphoribosyltransferase metabolized hypoxanthine to a purine nucleotide and inosine monophosphate with consumption of phosphoribosyl pyrophosphate [24,26,27,28]. These materials are recycled and get involved in the *de novo* pathway of purine metabolism. It is reported that a reutilization rate of 90% of hypoxanthine through the salvage pathway and conversion to inosine monophosphate [29]. These differential regulations of metabolism of hypoxanthine, xanthine, and uric acid may affect the difference of association between plasma XOR activities and purine metabolism-associated markers.

Renal urate transporter 1 expression is decreased in patients with alcohol-associated liver disease and viral hepatitis C [30]. The transcript and protein levels of hypoxanthine transporter, namely equilibrative nucleoside transporters [31] are changed in the liver during ischemia and reperfusion [32] although the membrane transport mechanism of xanthine is unknown [31]. Thus, purine metabolism-related markers are likely to be affected by liver dysfunction.

Activation of XOR causes activation of XO and NADH oxidase, which generates ROS, as mentioned above [1]. The close relationship between plasma XOR activities and the levels of d-ROMs, even in patients with liver diseases and controls, is reasonable because d-ROMs are a certain biomarker for quantification of the oxidative status.

The natural logarithmic value of XOR activity in plasma is reportedly correlated with hs-CRP in healthy volunteers [33], which is consistent with our finding that the plasma XOR activities were well correlated with hs-CRP values even in patients with liver diseases and controls. Plasma XOR activity may reflect not only subclinical [33] but also overt clinical inflammation in liver diseases.

Serum or hepatic XO has been evaluated in human liver diseases [10,11]. Serum XO was determined by competitive enzyme-linked immunosorbent assay (ELISA) in 12 controls, 17 patients with cirrhosis, 30 patients with chronic hepatitis, and 17 patients with cholestatic disorders [10]. The authors concluded that serum XO reflected the presence of cholestasis, which exhibited the highest serum XO levels [10]. The number of patients in each group was small, and the etiologies consisted of multiple diseases, especially cholestatic disorders in which cholangiocarcinoma was included [10]. Our results showed that plasma XOR activity might reflect serum ALT, γGTP, and ALP levels, but numerically, the strongest linear relationship was observed between plasma XOR activity and serum ALT levels in the study population. In PBC patients, plasma XOR activities showed a significant difference between subjects with normal and abnormal serum ALT values but not ALP or γGTP. Hepatic XO was determined spectrophotometrically in 12 normal liver tissue samples of metastatic liver tumors, 32 virus-related cirrhosis, 8 non-virus-related cirrhosis, and 4 HCC samples [11]. The results showed that hepatic XOR activities correlated with serum ALT levels in virus-related cirrhosis and were higher than those in controls, while hepatic XOR activities in non-virus-related cirrhosis and HCC were lower than those in controls [11]. Except for the close correlation with serum ALT levels, the hepatic XOR activity results were inconsistent with our findings, as mentioned above. Notably, hepatic XOR activity was below the detection limits in acute hepatitis B, which is totally different from our results. Although the exact reason is unknown, the small number of tissue samples and the difference in the specimen material might have caused the discrepancy. In our study, plasma XOR activity was strongly correlated with serum ALT levels, and the findings were consistent regardless of the liver etiology.

In addition to the study mentioned above [11], lower XDH mRNA expression in HCC tissues has been reported [34]. In our study, however, plasma XOR activity was higher in uncured patients with active HCC than in controls or cured HCC patients, which might indicate increased inflammation and augmentation of oxidative stress in HCC patients.

In subjects with NAFLD, a study showed that plasma XOR activity correlates with hepatic steatosis and serum uric acid levels by using the same measurement methods as ours [35], and serum XOR activity was significantly related to NAFLD by ELISA [36], which is consistent with our findings. In fact, plasma XOR activity was significantly correlated with BMI even in subjects with various liver diseases in our study (Figure 7), which supports the high plasma XOR activity in NAFLD [35,36] and was consistent with a report detailing the close correlation between plasma XOR activities and BMI in healthy volunteers [33].

Our study was limited by the relatively small number of patients with LC in some etiologies and HCC, the low number of patients in some groups, such as those with overlap syndrome, and the fact that there was only one patient with acute hepatitis. As the design of our study is cross-sectional, a cause-and-effect relationship between plasma XOR activities and biochemical parameters such as parameters of liver tests cannot be concluded. Similarly, as our study subjects were only Japanese, the results may not be generalized to all races.

Unfortunately, we could not register the patients with drug-induced liver injury (DILI) and herb-induced liver injury (HILI) in this study. The plausible reasons are that each attending physician may be concerned about several points as follows: (1) these diseases are diagnosed basically based on exclusive diagnosis and we often lose the chance to take blood at the peak of serum transaminase levels. Then, serum transaminase levels are likely to be close to the basal levels at scheduled date of blood collection. (2) As our study was performed in flagship hospitals, these patients tend to have several underline diseases such as nonhepatic cancers, severe inflammation diseases, or active autoimmune diseases, which may affect plasma XOR levels. (3) As plasma XOR levels may be affected based on the type of drugs or herbs, the results of plasma XOR levels may vary in patients with DILI and HILI.

In summary, plasma XOR activities were generally increased in liver diseases, especially in the active phase, and were closely correlated with liver transaminase levels, especially serum ALT levels, regardless of etiology and plasma xanthine levels. Our results indicated that high plasma XOR activity might reflect the active phase in various liver diseases. Longitudinal and interventional studies, such as the comparison between the acute phase and recovery phase in the same liver patients with not only Japanese but also other ethnicities, warrant further investigation.

## Figures and Tables

**Figure 1 biomedicines-11-01445-f001:**
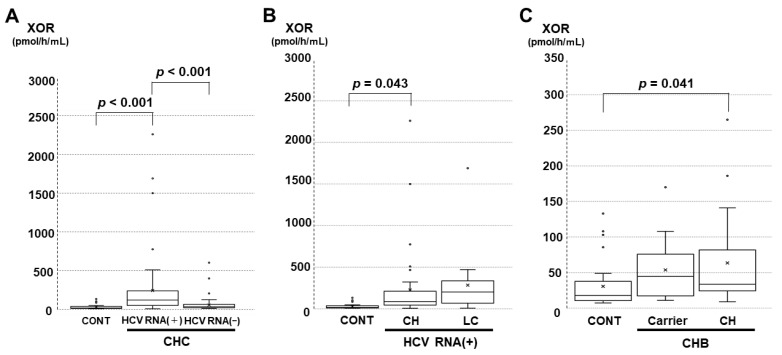
Comparisons of plasma xanthine oxidoreductase (XOR) activities among the control (CONT) and viral hepatitis groups under specified conditions. The comparisons were performed with the Kruskal‒Wallis test and Bonferroni’s multiple comparison test as post hoc tests. (**A**) Comparison among CONT, chronic hepatitis C (CHC) patients with serum hepatitis C virus (HCV) RNA positivity [HCV RNA (+)], and those with serum HCV RNA negativity [HCV RNA (−)]. (**B**) Comparison among CONT, chronic hepatitis C without liver cirrhosis (CH), and liver cirrhosis (LC) with HCV RNA (+). (**C**) Comparison among CONT and chronic hepatitis B (CHB) that includes inactive hepatitis B virus (HBV) carriers or asymptomatic HBV carriers (Carrier) and CH infected with HBV.

**Figure 2 biomedicines-11-01445-f002:**
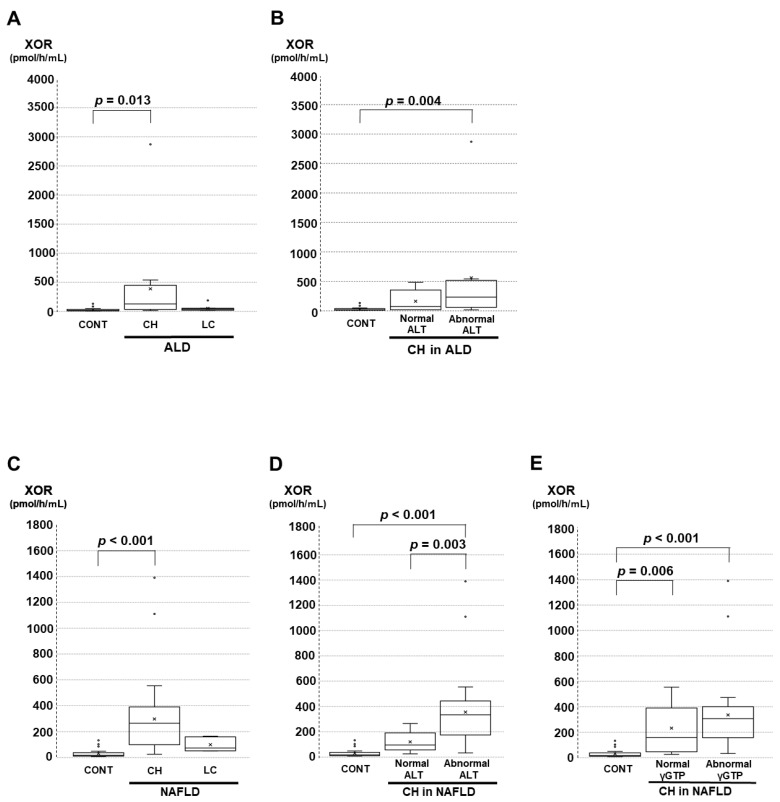
Comparisons of plasma XOR activities among the CONT and alcoholic liver disease (ALD) or nonalcoholic fatty liver disease (NAFLD) groups under specified conditions. The comparisons were performed with the Kruskal‒Wallis test and Bonferroni’s multiple comparison test as post hoc tests. (**A**) Comparison among CONT and ALD (CH and LC). (**B**) Comparison among CONT and CH in ALD with normal ALT levels and abnormal levels. (**C**) Comparison among CONT and NAFLD (CH and LC). (**D**) Comparison among CONT and CH in NAFLD with normal ALT levels and abnormal ALT levels. (**E**) Comparison among CONT and CH in NAFLD with normal and abnormal γGTP levels.

**Figure 3 biomedicines-11-01445-f003:**
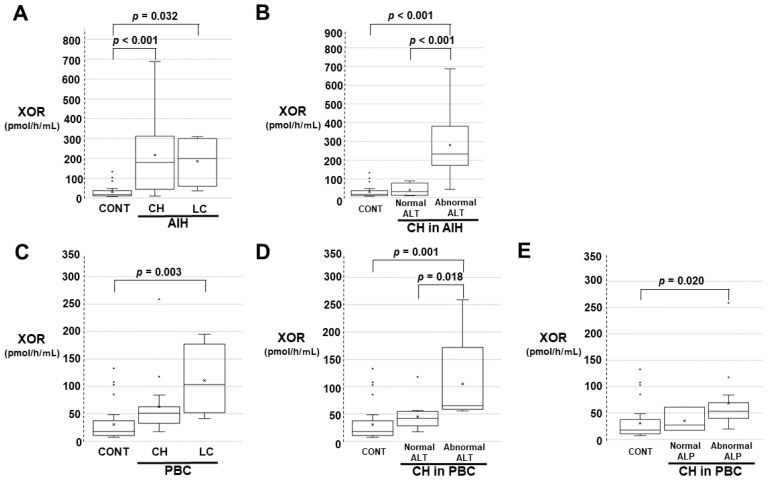
Comparison of plasma XOR activities among the CONT and autoimmune disease (AIH) or primary biliary cirrhosis (PBC) groups under specified conditions. The comparisons were performed with the Kruskal‒Wallis test and Bonferroni’s multiple comparison test as post hoc tests. (**A**) Comparison among CONT and AIH (CH and LC). (**B**) Comparison among CONT and CH in AIH with normal ALT levels and abnormal ALT levels. (**C**) Comparison among CONT and PBC (CH and LC). (**D**) Comparison among CONT and CH in PBC with normal ALT levels and abnormal ALT levels. (**E**) Comparison among CONT and CH in PBC with normal and abnormal ALP levels.

**Figure 4 biomedicines-11-01445-f004:**
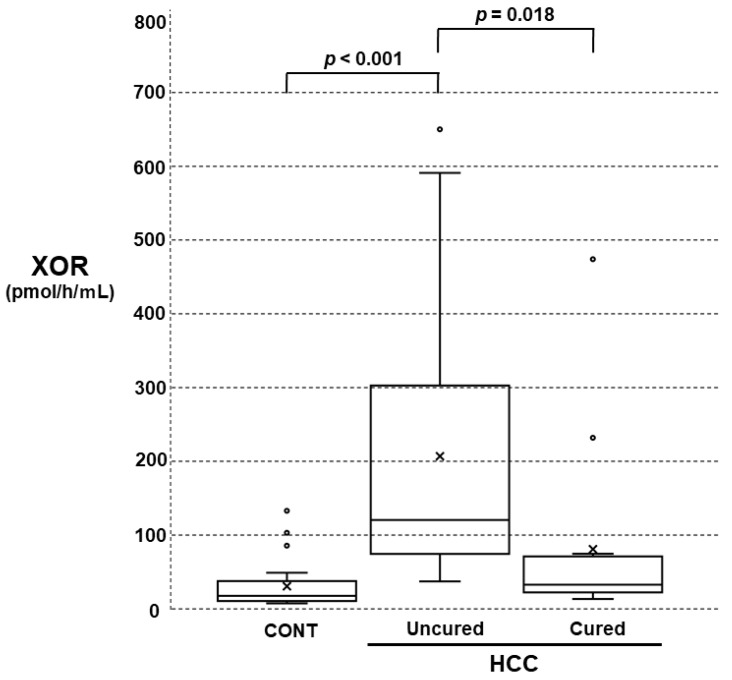
Comparison of plasma XOR activities among the CONT and hepatocellular carcinoma (HCC) (uncured and cured) groups. The comparisons were performed with the Kruskal‒Wallis test and Bonferroni’s multiple comparison test as post hoc tests.

**Figure 5 biomedicines-11-01445-f005:**
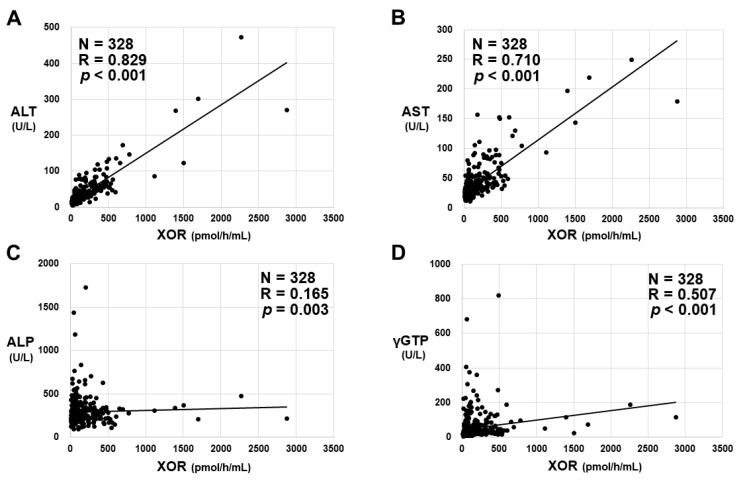
The relationship between plasma XOR activity and the levels of parameters of liver tests: (**A**) alanine aminotransferase (ALT), (**B**) aspartate aminotransferase (AST), (**C**) alkaline phosphatase (ALP), and (**D**) γ-glutamyl transpeptidase (γGTP). The correlation between two values was analyzed using Spearman’s rank correlation coefficient. N; the number of subjects, R; correlation coefficient.

**Figure 6 biomedicines-11-01445-f006:**
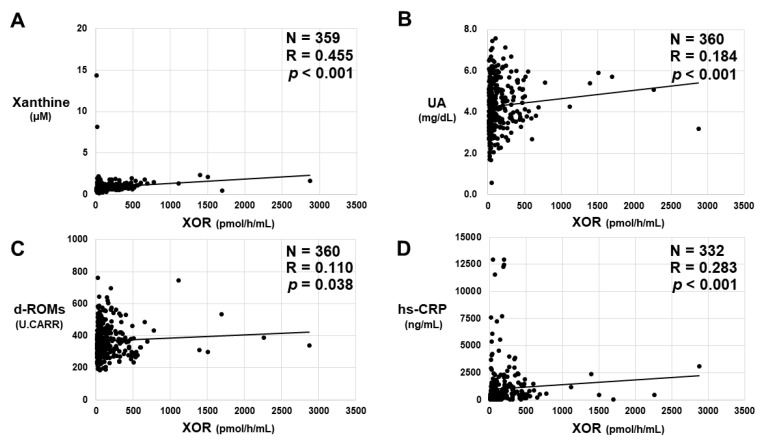
The relationship between plasma XOR activities and the values of the purine metabolism-related markers (**A**) xanthine and (**B**) uric acid (UA), the oxidative stress marker (**C**) derivatives of reactive oxygen metabolites (d-ROMs) and the inflammation marker (**D**) high-sensitivity C-reactive protein (hs-CRP). The correlation between two values was analyzed using Spearman’s rank correlation coefficient. N; the number of subjects, R; correlation coefficient.

**Figure 7 biomedicines-11-01445-f007:**
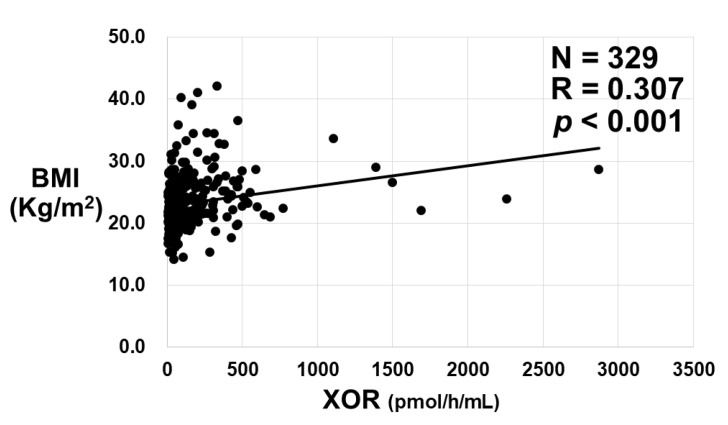
The relationship between plasma XOR activities and the values of body mass index (BMI). The correlation between two values was analyzed using Spearman’s rank correlation coefficient. N; the number of subjects, R; correlation coefficient.

**Table 1 biomedicines-11-01445-t001:** Characteristics of the study population. BMI: body mass index; HCV: hepatitis C virus; HIV: human immunodeficiency virus; IQR: interquartile range; NA: not available; PSC: primary biliary cholangitis.

Groups	Total	Patients	Controls
**Number of subjects**	361	329	32
**Age (years) median (IQR)**	66 (54–73)	67 (57–74)	39 (33–45)
**Gender (Male/Female)**	152/209	144/185	8/24
**BMI (kg/m^2^) median (IQR)**	NA	22.8 (20.7–25.5)	NA
**Etiology (Number)**			
Chronic hepatitis C		165	
Chronic hepatitis C with HIV infection		1	
Chronic hepatitis B		40	
Chronic hepatitis B with past HCV infection		2	
Acute hepatitis B		1	
Non-B non-C liver cirrhosis		3	
Nonalcoholic liver disease (NAFLD)		46	
Alcoholic liver disease (ALD)		26	
Autoimmune hepatitis (AIH)		19	
Primary biliary cholangitis (PBC)		22	
Overlap syndrome (AIH + PBC)		3	
Overlap syndrome (AIH + PSC)		1	
**Stage of liver disease (Number)**			
Acute hepatitis		1	
Chronic liver disease without cirrhosis		255	
Liver cirrhosis (Child–Pugh score A/B/C)		73 (57/15/1)	
Hepatocellular carcinoma		28	
**Liver transplantation**		0	
**Complications (Number)**			
Diabetes mellitus		77	
Hypertension		156	
Hyperuricemia		32	
Hyperlipidemia		119	

**Table 2 biomedicines-11-01445-t002:** Plasma XOR activities due to etiology of liver disease. Statistical significance between the control group and each other group is listed in the table. The comparison between the control group and the groups in which the number of subjects were less than three was not performed. AIH: autoimmune disease; ALD: alcoholic liver disease; HIV: human immunodeficiency virus; NA: not applicable; NAFLD: nonalcoholic fatty liver disease; NBNC: non-B non-C; PBC: primary biliary disease. * Chronic hepatitis B with past HCV infection.

Control/Disease	N	Minimum	1st Quartile	Median	3rd Quartile	Maximum	*p*-Value
Controls	32	7.3	10.7	17.7	35.6	133	NA
Chronic hepatitis C	165	6.84	24.0	47.7	112	2260	<0.001
Chronic hepatitis B	40	8.9	26.3	38.7	74.7	265	<0.001
Chronic hepatitis B/C *	2	12.2	NA	NA	NA	48.5	NA
Chronic hepatitis C/HIV	1			121			NA
NAFLD	46	25.7	92	217	367.8	1390	<0.001
ALD	26	13.3	21.7	52	277.8	2870	<0.001
PBC	22	17.4	41.7	53.4	79.4	259	<0.001
AIH	19	9.9	66.8	180	290	688	<0.001
Overlap syndrome (AIH/PBC)	3	39.3	NA	194	NA	430	0.009
Overlap syndrome (AIH/PSC)	1			64.5			NA
NBNC cirrhosis	3	59.8	NA	82.1	NA	139	0.013
Acute hepatitis B	1			>6670			NA

## Data Availability

The analyzed datasets in the study are available from the corresponding author on reasonable request.

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
