# Peer review of "A Newly Developed Method-Based Xanthine Oxidoreductase Activities in Various Human Liver Diseases"

_biomedicines, 2023, doi:10.3390/biomedicines11051445_

Round 1
Reviewer 1 Report
I have very little to suggest - the authors seem to have performed a well organised and carefully analysed study. The findings seem to be largely consistent with the existing literature, which they have cited. I would maybe suggest rescaling the axes in Fig 2 for easier visual comparison between samples, but otherwise nothing to add.
Author Response
Responses to Reviewer 1
Comments and Suggestions for Authors
I have very little to suggest - the authors seem to have performed a well organised and carefully analysed study. The findings seem to be largely consistent with the existing literature, which they have cited. I would maybe suggest rescaling the axes in Fig 2 for easier visual comparison between samples, but otherwise nothing to add.
Thank you very much for your comments. We rescaled the axes in Fig. 2 for easier visual comparison between samples. Please see Pages 7 and 8.
Reviewer 2 Report
This paper, showing that XOR activity reflects liver disease in humans, is a useful finding, although the detailed mechanism including the involvement of ROS, is expected to be elucidated in the future.
(1) A high correlation was observed between plasma XOR activity and ALT. From the author's discussion, I understand that this may be due to the fact that XOR and ALT are released into plasma during hepatocellular injury. However, does the XOR activity measured in this study reflect the amount of XOR protein expression secreted into plasma? XOR activity is measured from the isotope-labeled xanthine, uric acid. So, is it possible that endogenous hypoxanthine, xanthine, uric acid which are originally present in plasma, could competitively inhibit the XOR activity? It is concerned that XOR activity may not reflect protein levels.
(2) In Fig. 6, it says that positive correlations were observed for A, B, C, and D, but this cannot be determined from the figures. The regression lines should be shown in each figure.
(3) Page 10, 11; The authors state that the figure in which no correlation was observed is "data not shown". In particular, hypoxanthine is a substrate of XOR and should be presented as a figure even if it did not correlate with XOR activity. The relationship between hypoxanthine (the XOR substrate), xanthine (the substrate and metabolite), uric acid (the metabolite), and XOR activity should be discussed in an integrated manner. Is it possible that these concentrations may fluctuate due to changes in transporters such as the uric acid transporter during liver disease?
Author Response
Responses to Reviewer 2
Comments and Suggestions for Authors
This paper, showing that XOR activity reflects liver disease in humans, is a useful finding, although the detailed mechanism including the involvement of ROS, is expected to be elucidated in the future.
Thank you very much for your comments.
(1)A high correlation was observed between plasma XOR activity and ALT. From the author's discussion, I understand that this may be due to the fact that XOR and ALT are released into plasma during hepatocellular injury. However, does the XOR activity measured in this study reflect the amount of XOR protein expression secreted into plasma?
Plasma XOR activity seems to reflect XOR protein levels that are released from hepatocytes and bile duct epithelial cells during liver damage to some extent. However, a part of XOR protein was not reflected in plasma XOR activity. The reason is that XOR is composed of three domains that is Fe-S, FAD, and molybdopterin domains and molybdenum-containing enzyme can itself include a second-inactive form that is desulpho-XOR, in which a Mo=S grouping is replaced by Mo=O. Please see the text in the last paragraph on Page 13.
XOR activity is measured from the isotope-labeled xanthine, uric acid. So, is it possible that endogenous hypoxanthine, xanthine, uric acid which are originally present in plasma, could competitively inhibit the XOR activity?
To prevent these possibilities, the samples were purified on a Sephadex G-25 column in our study as with the previous reports. Please see the text in the first paragraph on Page 4 and the last paragraph on Page 13.
It is concerned that XOR activity may not reflect protein levels.
We answered your comment just as mentioned above.
(2) In Fig. 6, it says that positive correlations were observed for A, B, C, and D, but this cannot be determined from the figures. The regression lines should be shown in each figure.
We remade the figures and supplemental figures and added the regression lines in each graph. Please see each graph.
(3) Page 10, 11; The authors state that the figure in which no correlation was observed is "data not shown". In particular, hypoxanthine is a substrate of XOR and should be presented as a figure even if it did not correlate with XOR activity.
We made the supplemental figures 1 and 2 regarding the association between the various parameters and plasma XOR activity in case that these parameters were not significantly associated with plasma XOR activity. In addition, we made Figure 7 in which the significant association between BMI and plasma XOR activity was shown.
The relationship between hypoxanthine (the XOR substrate), xanthine (the substrate and metabolite), uric acid (the metabolite), and XOR activity should be discussed in an integrated manner.
Actually, we have already written these relationships in the “Introduction” and “Discussion” section in the original manuscript. However, we added the sentences in more detail and explain the salvage pathway that is hard to understand in the original manuscript. Please see the text in the second paragraph on Page 14.
Is it possible that these concentrations may fluctuate due to changes in transporters such as the uric acid transporter during liver disease?
The expressions of renal urate transporter 1 and liver equilibrative nucleoside transporters are altered in liver diseases. We added the several sentences in the third paragraph on Page 14.
Reviewer 3 Report
Interesting clinical study. I have only a few suggestions.
Minor points:
1. Table 1, can you please include data on DILI and HILI, assessed for causality by the updated RUCAM? If that is not possible, explain that in the text.
2. Consider that ALT is not a parameter of liver function but of liver tests, Please correct.
3. Introduction first line: 2x "that", replace one by: , which
Author Response
Responses to Reviewer 3
Comments and Suggestions for Authors
Interesting clinical study. I have only a few suggestions.
Thank you very much for your comments.
Minor points:
- Table 1, can you please include data on DILI and HILI, assessed for causality by the updated RUCAM? If that is not possible, explain that in the text.
Unfortunately, we could not register the patients with drug-induced liver injury (DILI) and herb-induced liver injury (HILI) in this study. The plausible reasons are described in the fifth paragraph on Page 15.
- Consider that ALT is not a parameter of liver function but of liver tests, Please correct.
We corrected the corresponding phrases throughout the text.
- Introduction first line: 2x "that", replace one by: , which
We corrected the corresponding phrase in the first line in the “Introduction” section.
Round 2
Reviewer 2 Report
The manuscript is well-revised according to the reviewer's comments.